# Hetero-Porous, High-Surface Area Green Carbon Aerogels for the Next-Generation Energy Storage Applications

**DOI:** 10.3390/nano11030653

**Published:** 2021-03-08

**Authors:** Bony Thomas, Shiyu Geng, Mohini Sain, Kristiina Oksman

**Affiliations:** 1Division of Materials Science, Department of Engineering Sciences and Mathematics, Luleå University of Technology, SE-971 87 Luleå, Sweden; bony.thomas@ltu.se (B.T.); shiyu.geng@ltu.se (S.G.); m.sain@utoronto.ca (M.S.); 2Mechanical & Industrial Engineering (MIE), University of Toronto, Toronto, ON M5S 3G8, Canada

**Keywords:** carbon aerogels, lignin, cellulose nanofibers, electrochemical properties, energy storage

## Abstract

Various carbon materials have been developed for energy storage applications to address the increasing energy demand in the world. However, the environmentally friendly, renewable, and nontoxic bio-based carbon resources have not been extensively investigated towards high-performance energy storage materials. Here, we report an anisotropic, hetero-porous, high-surface area carbon aerogel prepared from renewable resources achieving an excellent electrical double-layer capacitance. Two different green, abundant, and carbon-rich lignins which can be extracted from various biomasses, have been selected as raw materials, i.e., kraft and soda lignins, resulting in clearly distinct physical, structural as well as electrochemical characteristics of the carbon aerogels after carbonization. The obtained green carbon aerogel based on kraft lignin not only demonstrates a competitive specific capacitance as high as 163 F g^−1^ and energy density of 5.67 Wh kg^−1^ at a power density of 50 W kg^−1^ when assembled as a two-electrode symmetric supercapacitor, but also shows outstanding compressive mechanical properties. This reveals the great potential of the carbon aerogels developed in this study for the next-generation energy storage applications requiring green and renewable resources, lightweight, robust storage ability, and reliable mechanical integrity.

## 1. Introduction

Development of green and sustainable energy storage devices has been the focus of various researchers around the world in order to address the environmental challenges, energy crisis and rural development. Most of the current energy storage applications make use of lithium-ion batteries (LIBs) and supercapacitors (SCs) as the primary candidates to store energy. LIBs possess high energy density, but they exhibit low power density and poor cycle lives, whereas SCs exhibit high power density, fast charging/discharging, and long cycle lives, which make them more attractive for energy storage applications. SCs are generally classified into electrochemical double-layer capacitors (EDLCs) and pseudo capacitors based on the mechanism of their charge storage [1,2]. In EDLCs, charge storage is based on nonfaradaic electrostatic interactions at the electrode/electrolyte interface, while in pseudo capacitors the mechanism is based on faradaic redox reactions. Developing porous carbon-based EDLCs that exhibit superior electrochemical performances has always been the objective of researchers and various carbon-based electrode materials ranging from 0D to 3D—materials such as fullerenes, activated carbon, carbon nanotube, graphene, and metal organic framework derived carbon, etc., have been explored. Although these carbon electrode materials display excellent electrochemical performances, their complex synthesis routes, high cost of raw materials, chances of harsh environmental conditions and contaminations are the potential disadvantages which have attracted the use of bio-based precursor materials for the synthesis of porous carbon materials [1,2,3,4].

Renewable and naturally abundant biomass resources such as lignocellulosic materials are found to be the potential candidates for developing new energy storage materials [5,6]. Lignin, being the second most abundant aromatic polymer on the earth, possesses many attractive properties such as high carbon content, high thermal stability, low cost and eco-friendliness [5,7,8]. These properties have led to lignin being the primary candidate for developing bio-based carbon materials that can be used for energy storage applications [9]. Lignin is obtained as a byproduct from the paper and pulp industry and approximately 98% of the total production of approximately 70 million tons is used for energy recovery—i.e., to produce heat, power and biofuels [10]. Efforts have been made for the selective transformation of lignin into value added chemicals such as oligomers, phenols, aldehydes, ketones, acids, and esters. The rest of the material is utilized for low value products such as concrete additives, land fillings, dispersants and surfactants [11,12].

Thermochemical conversion methods such as pyrolysis followed by chemical activation [13,14,15,16] and hydrothermal carbonization followed by activation [17,18,19] have been investigated for the preparation of various porous carbon materials from different lignin precursors. Activated carbon materials from lignin possess many characteristics such as high specific surface areas (SSAs), hierarchical pore structures, high porosity, low density and high electrical conductivity, which make them suitable materials for supercapacitor (SC) electrodes [20]. Requirement for free standing, mechanically stable and lightweight carbon electrodes for energy storage applications has led to the development of biomass derived carbon aerogels (CAs) [9,20,21,22,23]. An SC electrode could be considered as ideal, when its microstructure contains pores of different size scales, i.e., macropores (>50 nm), which act as ion-buffering reservoir, mesopores (2–50 nm) for ion transportation, and micropores (<2 nm), which enhance the charge storage [21,24]. In addition to fact that CAs possess hierarchical pore structures (HPCs), their microstructures can be tuned in such a way that they can achieve higher electrochemical properties [25]. Conventionally, CAs are prepared from pyrolysis of organic polymer frameworks produced using sol-gel technique [26,27] or template assisted techniques [28,29]. These conventional methods are characterized by some difficulties such as (1) lack of enough mesopores in activated carbons for efficient ion transportation (even though they have very high SSAs) and mismatch between pore size and size of electrolyte ions; (2) highly ordered pores with narrow pore size distribution in the case of templated porous carbons; (3) high-cost and complex procedure needed for the preparation of nanomaterials (such as graphene [30,31], carbon nanotubes [32,33], and metal nanoparticles [34,35]) and incorporated Cas; (4) poor mechanical properties [36]. Research has been performed over the years to overcome these difficulties and in one of our previous works [9] multifunctional anisotropic CAs were prepared from kraft lignin (KL) and TEMPO-oxidized cellulose nanofibers (TOCNFs) which acted as a sacrificial template during the carbonization process, generating micropores and mesopores in the final carbon aerogel structure. The use of ice-templating technique helped to generate the macropores in the direction of growth of ice crystals which were retained in the material after freeze drying [37,38]. Thus, obtained hierarchically porous anisotropic carbon aerogels [9] showed a specific capacitance of 124 F g^−1^ at a current density of 0.2 A g^−1^, indicating the possibilities for exploring lignin/CNF as a potential carbon source for the preparation of supercapacitor electrodes. Despite the huge possibilities of lignin as a potential source of various carbon materials, more research must be carried out to achieve the requirements for a high-performance supercapacitor electrode from lignin-based carbon. The literature shows that electrode materials possessing porous structures with appropriate pore sizes of approximately 5 nm are preferable for electrolyte ion transportation, which helps achieve the best electrochemical performance [39,40]. Therefore, in this study, we used mechanically separated cellulose nanofibers (CNFs) along with two different types of lignin, i.e., kraft lignin (KL) and soda lignin (SL), to prepare CAs for energy storage applications. Compared to our previous study, CNFs have larger diameters than TOCNFs, so it is possibile to generate pores with of more suitable sizes when acting as the sacrificial template during carbonization. Moreover, we focus on the comparison of the two types of lignins and investigate their effects on the structure and electrochemical performance of the derived CAs. Through this study, we further explore the possibility to tune the pore morphology by the appropriate selection of the type of CNFs and lignin in the carbon aerogel precursors, aiming for developing green carbon electrodes from renewable resources with superior energy-storage capacitance. To the best of our knowledge, there is no study with a similar focus has been published in the literature.

## 2. Experimental

### 2.1. Materials

Kraft lignin (KL) (low sulfonate content, Mw of approximately 10,000) purchased from Sigma-Aldrich (St. Louis, MO, USA), soda lignin (SL) (trade name: Protobind 2000, GreenValue Enterprises LLC, Media, PA, USA) and mechanically fibrillated cellulose nanofibers (CNFs) with approximate diameters of 15 nm, were prepared according to the procedure reported by Berglund et al. [41] from hardwood birch (Betula pendula) chips supplied by Natural Resources Institute, Luke, Finland. Sodium hydroxide (NaOH) and potassium hydroxide (KOH) pellets were purchased from Merck KGaA, Darmstadt, Germany.

### 2.2. Materials Preparation

Lignin/CNF suspensions of 2.4 wt.% solid content, containing KL and mechanically fibrillated CNFs in the ratios, 60:40, 70:30, 80:20 and 88:12 were prepared by mechanically mixing the KL powder, aqueous CNF suspension of 1 wt.% solid content. Distilled water was added additionally in required amounts to maintain the ratio of lignin to CNF and the final solid content of lignin/CNF suspensions. For SL/CNF suspensions (60:40, 70:30, 80:20, 88:12 and same solid content 2.4 wt.% as the KL/CNF), the procedure was somewhat different, the SL powder was mixed with distilled water and dissolved using 2 M NaOH solution which was added dropwise into mixture and mixed for 6 h, at 60 °C keeping the pH value between 9 and 10. This lignin solution was then mixed with CNFs (1 wt.%) and mixed properly until a homogeneous suspension of the SL/CNF was obtained. The difference is because the KL is soluble with water while SL need to be solubilized using NaOH. This also meant that it was not possible to prepare the suspension with lowest lignin concentration (60%) using 1 wt.% CNF suspension keeping the same solid content as the others. To overcome this issue, the concentration of the CNF suspension was 2.3% in order to have enough distilled water to dissolve SL properly. After measuring the suspension properties such as viscosity and conductivity, all the suspensions were transferred to the refrigerator at 4 °C for 12 h. Ice-templating which was followed by freeze drying was used for the preparation of lignin/CNF aerogels. In the ice-templating, the lignin/CNF suspensions were poured into the Teflon molds which were attached on the top of the copper rod whose bottom end was dipped directly in the liquid nitrogen bath. The temperature at the top end of the copper rod was controlled by heating at the rate of 10 °C/min using a temperature control unit. Because of the temperature gradient between the suspension and the copper rod, unidirectional freezing from bottom to top can be achieved. Frozen samples were kept in the freezer at −20 °C for at least 2 h. Freeze drying was performed in an Alpha 1–2 LD plus freeze dryer (Martin Christ GmbH, Osterode, Germany) where fully frozen samples were subjected to a pressure of 1.00 mbar for 72 h. The shelf temperature was maintained at 30 °C during the entire freeze-drying process.

In order to prepare carbonization of the CAs, a Nabertherm RHTC-230/15 horizontal tube furnace (Nabertherm GmbH, Lilienthal, Germany) in nitrogen (N_2_) atmosphere was used. Abbreviations for the names of different CAs and their compositions are listed in Table 1.

In the first step, the aerogels were heated from room temperature to 100 °C at a rate of 5 °C/min and maintained at 100 °C for one hour to remove moisture. In the second step, the temperature was increased from 100 to 400 °C at a rate of 5 °C/min and kept isothermally at 400 °C for one hour. Further heating was carried out from 400 to 1000 °C at a rate of 5 °C/min followed by isothermal holding at 1000 °C for 1 h. After carbonization, the samples were cooled to room temperature and taken out carefully for further characterizations.

### 2.3. Supercapacitor Assembly

In order to fabricate the SC for two-electrode measurements, CAs were directly used as electrodes without any binder or conductive additive in symmetric configuration. The electrolyte used was 6 M KOH. Whatman filter paper (Grade 1, GE Healthcare, Machelen, Belgium, pore size: 22 µm) was used as the separator and Inconel 600 superalloy as current collectors.

### 2.4. Characterization

The viscosity of lignin/CNF suspensions was measured using a Vibro Viscometer SV-10, (A&D Company, Ltd., Tokyo, Japan), at a constant shear rate. The sensor plates had a periodic sinusoidal vibrational motion with a frequency of 30 Hz and amplitude <1 mm. The pH of lignin-CNF suspensions was measured using a Hanna pH meter (HannaNorden AB, Kungsbacka, Sweden) which is capable of measuring pH in the 0 to 14 range, with 0.01 resolution. A Mettler Toledo conductivity meter (Schwerzenbach, Switzerland) with a measuring range from 0.01 µS/cm to 500 mS/cm was used for measuring suspension conductivity. All these measurements were repeated three times and the average values are reported.

Thermal behaviors of KL, SL and CNF were studied separately using a thermogravimetric analyzer (TGA), TA Instruments TGA-Q500 (New Castle, DE, USA). The analysis was performed at a heating rate of 10 °C/min from room temperature to 1000 °C under nitrogen (N_2_) atmosphere. In order to confirm the actual amount of carbon in the Cas, TGA was performed under air atmosphere at a rate of 5 °C/min from room temperature to 1000 °C.

Raman spectroscopic measurements were carried out using a Bruker Senterra Raman microscope (Bruker corp., Billerica, MA, USA). The spectra were recorded from 50 to 4500 cm^−1^ using a laser power of 2 mW and a magnification of 20×. The XRD diffractograms of the KL, SL and CAs were recorded by PANalytical EMPYREAN equipment (Malvern Instruments, Malvern, UK) using a Cu kα radiation and the equipment was fitted with a PixCel3D detector and a graphite monochromator.

Carbon yield was calculated using Equation (1), where mass of the aerogel before and after carbonization was measured for the determination of the yield.
(1)$$\mathrm{Carbon}\;\mathrm{yield}=\frac{\mathrm{mass}\;\mathrm{of}\;\mathrm{carbon}\;\mathrm{aerogel}\;}{\mathrm{mass}\;\mathrm{of}\;\mathrm{precursor}\;\mathrm{aerogel}\;}\times 100$$

Bulk density of the CAs was estimated by dividing the mass by the bulk volume of the aerogels. Bulk volume was estimated by measuring the physical dimensions of the CAs. Porosity was measured using Equation (2).
(2)$$\mathrm{Porosity}\;=\;1-\frac{\rho_{b}}{\rho_{c}}$$
where *ρ_b_* is the bulk density of CA and *ρ_c_* is the density of amorphous carbon (2.26 g/cm^3^) [42]. Shrinkage in volume of aerogels occurred during the carbonization process was estimated by measuring the volume before carbonization (*V_i_*) and volume after carbonization (*V_f_*) using Equation (3).
(3)$$Volume\;shrinkage=\;\frac{V_{i}-V_{f}}{V_{i}}\;\times 100$$

Structure and morphology of the CAs were investigated using scanning electron microscopy (SEM) (JSM-IT300, JEOL, Tokyo, Japan). Samples were prepared by peeling off the top surface using a tape so that structure could be preserved without any deformations. Morphology of CAs was analyzed in two directions, parallel and perpendicular to the direction of ice-templating, with an acceleration voltage of 20 kV. Elemental composition of CAs was analyzed using energy dispersive X-ray spectroscopy (EDX), using the same SEM instrument equipped with a silicon drift detector (Oxford X-MaxN 50 mm^2^, Oxford Instruments, Abingdon, UK).

SSA, average pore size and pore volume of CAs were measured using a Brunauer–Emmett–Teller (BET) analyzer Micromeritics Gemini VII 2390a (Micromeritics Instrument Corporation, Norcross, GA, USA) at 77 K. The CAs were degassed using a Micromeritics FlowPrep060 sample degassing system at 300 °C for 3 h prior to the BET measurement to remove the already adsorbed moisture. Micropore area was calculated using a standard t-plot method and total pore volume was determined from the amount of N_2_ adsorbed at a relative pressure P/P_0_ = 0.99. Pore size distributions were calculated by Nonlocal Density Functional Theory (NLDFT) based on a slit pore model, using the built-in software.

Electrochemical properties of CAs were measured using a Princeton Applied Research VerstaSTAT 3 Potentiostat/Galvanostat (AMETEK Scientific Instruments, Wokingham, UK). A two-electrode method was followed for the electrochemical measurements. In order to analyze the rate capabilities of the electrodes, cyclic voltammetry measurements (CV) were carried out at different scan rates between 2 and 100 mV s^−1^ in the potential range from 0 to 1 V. Further investigation on the rate capabilities was investigated by means of the galvanostatic charge discharge method (GCD) at different current densities in the range from 0.1 to 1 Ag^−1^. Specific capacitances obtained from CV tests can be calculated using Equation (4),
(4)$$C=\frac{2}{mv\left(V_{2}-V_{1}\right)}\int_{V_{1}}^{V_{2}}IdV$$
where *C* (F g^−1^) is the specific capacitance; *m* (mg) is the mass of active materials loaded in the working electrode; *v* (V s^−1^) is the scan rate; *I* (A) is the discharge current; *V*_2_ and *V*_1_ (V) are high and low potential limits of the CV tests. Specific capacitance from GCD measurements can be calculated using Equation (5).
(5)$$C=\;\frac{4I∆t}{m∆V}$$
where *C* (F g^−1^) is the specific capacitance; *I* (A) is the discharge current; ∆*t* (s) is the discharge time; ∆*V* (V) is the potential window; *m* (mg) is the total mass of electrode material.

Specific energy density (*E*) and specific power density (*P*) derived from galvanostatic tests can be calculated from Equations (6) and (7).
(6)$$E=\;\frac{C∆V^{2}}{2*4*3.6}$$
(7)$$P=\frac{E}{∆t}$$
where *E* (Wh kg^−1^) is the average energy density; *C* (F g^−1^) is the specific capacitance; ∆*V* (V) is the potential window; *P* (W kg^−1^) is the average power density and ∆*t* (s) is the discharge time. Electrochemical impedance spectroscopy (EIS) measurements were carried out in the frequency range of 10^−2^ to 10^5^ Hz to determine the resistance of the carbon aerogels. Cyclic stability of KLCA60 was analyzed using three electrode cell kit (Pine Research Instrumentation, Durham, NC, USA) connected to the Princeton Applied Research VersaSTAT 3 potentiostat/galvanostat. A platinum electrode was used as counter electrode and an Ag/AgCl electrode worked as the reference electrode. The electrolyte used was 1 M H_2_SO_4_.

Mechanical behaviors of the CAs were examined using a Q800 Dynamic mechanical thermal analyzer (TA Instruments, New Castle, DE, USA) in compression mode and in axial and radial directions. Measurements were carried out after equilibrating at 303 K with a preload of 0.03 N and an initial strain of 0.5% was also applied. Strain rate was fixed at 5%/min. The slope of the initial linear portion of the stress–strain curve was used for the determination of elastic modulus.

## 3. Results and Discussions

Kraft lignin-based carbon aerogels (KLCAs) and soda lignin-based carbon aerogels (SLCAs) were prepared by following the preparation steps, as showed schematically in Figure 1a. Lignin/CNF suspensions were prepared by dissolving lignin in distilled water (60, 70, 80 and 88 wt.% of the total dry weight) and then mixed with CNF. These suspensions were ice-templated and freeze-dried to obtain lignin/CNF aerogels which were carbonized at 1000 °C to obtain CAs.

Properties of the lignin/CNF suspensions such as viscosity or pH play vital roles in the ice-templating process by changing the ice growth velocity and change in the crystal morphology [43] which can influence the pore structure of lignin/CNF aerogels. SL Viscosity values of both KL and based suspensions (total solid content for both types of suspensions was 2.4 wt.%) were increased by increasing the percentage of CNFs in the suspension (from 2.5 to 252 mPa·s for KL and 4.5 to 210 mPa·s for SL), as shown in Figure 2a. 

Thermal behaviors of the different types of lignins and CNFs were analyzed using TGA and the curve is plotted as a function of temperature and shown in Figure 2b. A derivative thermogravimetric (DTG) curve has been represented in Appendix A. It has been observed that for all the three materials, major degradation happened in the temperature range between 200 and 400 °C. The major observations from the TGA are listed in Table 2.

The TGA curve for cellulose nanofibers showed a degradation peak at 249 °C which indicates the presence of minor amounts hemicellulose present in the switchable ionic liquid (SIL) treated cellulose nanofibers [41]. The maximum degradation rate of cellulose nanofibers occurred at 317 °C, leaving 24% char residue at 1000 °C. KL started to degrade at 260 °C and the degradation rate was the highest at 315 °C and it finally reached a char residue of 45% at 1000 °C. SL started degrading at an even lower temperature of 161 °C and the degradation reached a maximum at 357 °C, leaving 30% of char residue at 1000 °C. Degradation of lignin in the temperature range 250 to 350 °C corresponds to the cleavage of β-O-4 linkages [44]. This weight loss for lignins at around 300–400 °C mainly occurred due to the cleaving of alkyl (C-C) and methoxy groups, releasing methanol and phenolic compounds. Above 600 °C, decarbonylation of alkyl side chains happens along with large reduction in functional groups, leading to the formation of amorphous carbon [45]. The TGA suggests that the cellulose nanofibers can act as potential sacrificial templates during carbonization (as shown schematically in Figure 1), leaving a lower amount of residue and generating porous microstructures in KLCAs [46]. However, in the case of SLCAs, both SL and CNF had similar solid residues, indicating the inefficiency of CNFs as sacrificial templates, hence the possibility of them generating lower number of sacrificial pores as compared to KL/CNF systems. After the ice-templating, freeze drying and carbonization, the obtained KLCAs showed a carbon yield between 30% (KLCA60) and 45% (KLCA88), while for the SLCAs it ranged between 31% (SLCA60) and 36% (SLCA88), as shown in Appendix A, which is also consistent with the TGA results. The pH and conductivity values of lignin/CNF suspensions are listed in Table 3. The pH was in the range between 8.5 and 9.5 for all suspensions and the conductivity was between 1.5 and 3.5 mS cm^−1^.

Physical properties of CAs such as bulk density, bulk porosity, and volume shrinkage for all the aerogels are provided in Table 3. Bulk density of KLCAs was increased from 0.046 (KLCA60) to 0.061 g cm^−3^ (KLCA88), while that of SLCAs was increased from 0.024 (SLCA60) to 0.033 g cm^−3^ (SLCA88). Thus, both types of aerogels exhibited increases in bulk densities with increasing lignin contents and the values were higher for KLCAs than SLCAs. This can be correlated with the high carbon yield of KL compared to SL as discussed before. All KLCAs were around 97% porous while the porosity was around 98% for SLCAs. Volume shrinkage during carbonization ranged between 66% and 71% for KLCAs, whereas it was between 69% and 73% for SLCAs. A large reduction in the mass of lignin during the carbonization process as suggested by the TGA is the reason behind the volume shrinkage [47].

Raman spectra of KLCAs and SLCAs are represented in Figure 2c,d, respectively. Both KLCAs and SLCAs showed characteristic bands for carbon materials at 1582 (G-band) and at 1330 cm^−1^ (D-band). The G-band is due to the vibrations of sp^2^ bonded carbon atoms in the 2D hexagonal lattice and the D-band represents the in-plane terminated disordered tangling bonds in graphite. In both KLCAs and SLCAs, the ratio of intensity of the G-band to D-band (I_G_/I_D_) is more than 1, indicating the presence of partially graphitized carbon structures formed as a result of rearrangement of the carbon structure during the carbonization process, which is advantageous for making superior supercapacitor (SC) electrodes [48]. XRD patterns of KL and SL were recorded and represented in Figure 2e. In the case of KL, a diffraction peak was observed at 2θ = 18.7°, while for SL the peak was located at 2θ = 21.6°. After carbonization, both CAKL60 and CASL60 had broad peaks located at 2θ = 21° and 2θ = 44°, as shown in Figure 2f, which represent (002) and (101) planes of partially graphitized carbon structures. Stronger diffraction peaks of SLCA60 compared to KLCA60 indicating the occurrence of more graphitization during the carbonization for SL-based CAs.

The anisotropic structures of CAs were investigated in cross-sectional and longitudinal directions using scanning electron microscopy (SEM) and the images are shown in Figure 3 and Figure 4. The microstructures of both types of CAs have 3D macroporous, honeycomb-like interconnected structures in their cross-sections (Figure 3), while well-defined channel-like pores were observed in the axial directions (Figure 4) which were formed during the unidirectional ice-templating. The advantage of having macropores is that they are helpful in reducing the electrolyte ion diffusion distance to the active sites and decreasing the ion transport resistance and thus facilitate the formation of more efficient electrical double layers [24]. The sizes of macropores were found to increase with a decrease in the amount of CNFs (comparing KLCA60 and KLCA88 cross-sections in Figure 3), which is attributed to the lower viscosity of the suspensions (Figure 2a) and is in accordance with the reported literature for ice-templated nanocellulose aerogels [49,50]. In suspensions with higher viscosities, lateral growth of ice crystals is restricted due to the higher shear stress developed, inducing more nucleation sites, hence resulting in the formation of smaller macropores [51]. In Figure 4, it is clear that with increase in the lignin content, the longitudinal channels become wider, distorted and thicker. CAs prepared from lignin/CNF aerogels with higher CNF contents showed pores on the channel walls, and their porosity was derived from both the sacrificial templating of CNF and also the evolution of volatiles during carbonization. On the other hand, KLCA88 showed (Figure 4d) thicker and relatively smooth walls without porosity. From examining the longitudinal channels of CA microstructures, it was found that KLCA88 has the thickest and smoother cell walls which could lead to the trapping and deposition of volatiles evolved during carbonization of lignin. This trapping and condensation of volatiles could block the micro- and mesopores and lead to the formation of denser carbon structures with lower specific surface areas. The microstructures of SLCAs showed similar tendencies which are shown in Appendix A.

Elemental compositions of the CAs were qualitatively investigated using SEM-EDX and the results are listed in Table 4. Overall, it has been observed that the carbon content increased, and oxygen content decreased with increased lignin content for KLCAs (except KLCA70 which showed slight variation from the tendency). Small amounts of sodium and sulfur were present in KLCAs which might be from the kraft pulping process. There was a decrease in the weight percentage (wt.%) of Na and S with increasing amount of KL in KLCAs. The high Na and S contents in KLCAs with higher CNFs could be due to the evolution of more CO, CH_4_ and CO_2_ utilizing the oxygen and hydrogen functionalities present in CNFs during the carbonization, resulting in relatively higher number of inorganics compared to carbon. SLCAs showed almost similar wt.% of carbon, confirming the formation of carbon without generating a large amount of micropores and mesopores by giving away carbonaceous gases such as CO and CO_2_ during carbonization. It can be observed that minor amounts of silicon (Si) and potassium (K) were also present in the SLCAs, which could be attributed to the impurities from the SL resources such as wheat straw. From TGA performed under an oxidative atmosphere (Appendix A), only 7 wt.% remained of ash (metal oxides of Na) for KLCA60 and only 3 wt.% of ash for SLCA60 (oxides of Na or silicates). These results are consistent with the EDX results (Table 4 and Appendix A) which confirms that CAs possess carbon as their major component and have only minor contents of other elements.

The SSA, average pore diameter and pore volume of the CAs were measured using Brunauer–Emmett–Teller (BET) analysis. Nitrogen (N_2_) adsorption isotherms for the KLCAs and SLCAs are represented in Figure 5a,b, respectively. For KLCAs, higher nitrogen absorption was observed until the use of KLCA80 and then started to decrease for KLCA88. The sharp increment in the slope of adsorption isotherms at relative low pressures (P/P_0_ < 0.1) indicates the presence of microspores (<2 nm) that are present in the materials. The steady increase in the adsorption at higher relative pressures shows the monolayer/multilayer adsorption of the nitrogen molecule in the mesopores (2–50 nm), while the adsorption at the higher relative pressures (P/P_0_ ~ 1) indicates the total pore volume from both the micro- and mesopores. The adsorption isotherms indicate the presence of hierarchical pore structures in the KLCAs which is beneficial for electrochemical performance [21]. Reduction in SSA from KLCA80 to KLCA88 can be explained from the differences in the microstructure. In KLCAs obtained from higher CNF content, the SSA is contributed from the sacrificial templating and also from the porosity generated by the evolution of volatiles from the lignin itself during the carbonization process. However, KLCA88 had smooth and the thickest cell walls (Figure 4d), which resulted in the trapping and deposition of the volatiles in the pores, causing clogging which resulted in the formation of nonporous and dense carbon structures with the lowest SSAs. In comparison with KLCAs, SLCAs showed lower N_2_ adsorption at all relative pressures, see Figure 5b, probably due to the absence of hierarchical pore structure which could result in inferior properties as SC electrodes. KLCAs showed clearly distinct behavior compared to SLCAs during the carbonization process, which is evident from the BET results tabulated in Table 5. Comparison between SSA of KLCAs and SLCAs is shown in Figure 5c. KLCAs showed higher SSAs, micropore areas and total pore volumes compared to SLCAs. The highest SSA was obtained for KLCA80 (715 m^2^/g), while SLCA80 showed a low SSA of 151 m^2^ g^−1^.

It can be observed from Figure 5d that the SSA and pore volume have a linear relationship with micropore area. In KLCAs, both SSAs and pore volumes were increased with increased lignin content, except for KLCA88. Appendix A shows the pore distribution curves for CAs and KLCA60 showed the highest peak at around 2.5 nm and broad peaks between 4 and 6 nm indicating the abundance of smaller mesopores which contribute to the pore volume. Samples SLCA60, SLCA70 and SLCA88 had no considerable microporosity, resulting in very low SSA and low pore volume contributions by mesopores (as suggested by Appendix A); hence, it is probable that they will exhibit low electrochemical performances. A similar tendency of low observable micropores and SSAs for carbonized SL particles has been reported by Köhnke et al. [52] and Lin et al. [53]. Compared to the previous studies for carbonization of SL particles, the current method of ice-templating followed by freeze drying and carbonization could achieve higher a SSA for the SLCA80 sample without using any chemical or physical activation techniques. However, similar amounts of solid residues obtained after the carbonization of SL and CNFs (as shown in TGA in Figure 2b) indicated that CNFs could not act as efficient sacrificial templates during carbonization, which was confirmed by the low SSAs for SLCAs as compared to KLCAs. Another reason for lower SSAs for SLCAs could be the tar formation which is always associated with lignin carbonization. In an experimental study to analyze the pyrolysis of lignin, Ferdous et al. [54] compared the tar formation behaviors of KL and Alcell lignin (AL) at a heating rate of 300 °C/h (same as the heating rate used for this study). They observed that KL produced only 2.7 wt.% of tar while AL produced 14.1 wt.% [54]. Similar to this observation, carbonization of SL might also produce high amount of tar which are not easily decomposed and remained in the CAs, thereby blocking the pores available in the materials. From a chemical structural point of view, the low SSA can be attributed to the high syringyl to guaiacyl ratio (S/G ratio) of SL compared to KL. It has been observed from differential scanning calorimetry (DSC) (Appendix A) that the KL has glass transition temperature (T_g_) at 135 °C, while SL showed lower T_g_ at 80 °C, indicating the early stage softening of SL compared to KL. Rowlandson et al. reported the melting of lignin which can produce an intermediate melt region where viscosity is dependent on the S/G ratio [48]. A low S/G ratio lignin such as KL will have high melt viscosity and difficulty to flow because of the abundance of β-5 linkages which limit the molecular rotation, while a high S/G ratio SL will have a low melt viscosity and hence can flow inside the materials closing the pores generated during carbonization.

The electrochemical properties of the CAs were studied through cyclic voltammetry (CV) and galvanostatic charge discharge (GCD) measurements using a symmetric two-electrode system with 6 M KOH solution as electrolyte. CV measurements for SCs assembled using KLCAs and SLCAs were carried out at different scan rates and the results are plotted in Figure 6 and Appendix A. Capacitance values obtained from CV and GCD measurements are tabulated in Appendix A. Cyclic voltammograms of KLCA60-SC showed near rectangular shapes even at a scan rate of 100 mV s^−1^, indicating the good rate capability of the KLCA60 electrode. KLCA70-SC and KLCA80-SC showed rectangular CVs up to 20 mV s^−1^ but shifted to quasi rectangular shapes at higher scan rates of 50 and 100 mV s^−1^. These CV results revealed the perfect double-layer capacitive behavior of SCs made from KLCA electrodes. For KLCA88-SC, CV curves deviated from the rectangular shape and the current density values increased at higher potentials (0.85–1 V), indicating undesirable faradaic losses happening probably due to the onset of carbon electrode corrosion reactions [55]. Compared to KLCA-SCs, those with SLCA electrodes had much lower SSAs and hence were more resistant to the penetrating ions, resulting in CVs which are not rectangular in shape. SLCA80 samples showed better CVs among the SL-based carbon aerogels. During the charge discharge measurements, KLCAs showed symmetrical isosceles triangles as shown in Figure 7, indicating their good capacitive behavior. In agreement with the results of CV, KLCA60-SC showed the highest discharging time and hence highest specific capacitance of 163.4 F g^−1^ at current density of 0.1 A g^−1^ and retained a specific capacitance of 96.8 F g^−1^ at 1 A g^−1^, as represented in Figure 8a. For other KLCAs (KLCA70—129 to 34 F g^−1^, KLCA80—78 to 44 F g^−1^, and KLCA88—106 to 34 F g^−1^ for current density of 0.1 to 1 A g^−1^, as shown in Figure 8a), the discharging time decreased, indicating a decrease in the electrochemical performance. Even though KLCA60 had a lower SSA compared to KLCA80 and KLCA70 (Table 5), the highest specific capacitance for KLCA60-SC could be based on the perfect combination of approximately equal amounts of microporous (51.3% of total SSA) and meso/macroporous (48.7% of total SSA; contribution from macropores is very negligible since the average pore diameter is 2.3 nm) surface area in the material, confirming the importance of achieving hierarchical pore morphology along with a considerably high mesoporous surface area. A similar inference was reported by Xu et al., where the specific capacitance was varied as a function of mesoporous surface area for lignin-based carbon aerogels [22]. Additionally, from the longitudinal microstructures of KLCAs (in Figure 4), KLCA60 has the lowest distance between adjacent walls, which is beneficial in making EDLCs compared to those with larger distances between cell walls such as in KLCA88.

Electrochemical impedance spectroscopy (EIS) has been carried out in the frequency range between 10^−2^ and 10^5^ Hz and the Nyquist plots for KLCA-SCs are shown in Figure 8b. All KLCAs showed very low equivalent series resistance (<1 Ω) and the KLCA60 sample showed less charge transfer resistance (19 Ω) compared to the others. The high charge transfer resistance can be attributed to the (i) electrolyte resistance in the pores of the electrodes, (ii) the contact resistance between electrode and current collector [21]. In these aerogels, increase in contact resistance could be due to the increase in size of macropores as a result of reduced viscosity of lignin/CNF suspensions at higher lignin contents. A similar tendency of the charge transfer resistance to increase with an increase in the size of macropores has been reported by Chang et al. [56].

The IR-drop observed in the GCD curves (Figure 6) agrees with the results obtained from the Nyquist plots. However, the nearly vertical lines in the EIS (as seen in Figure 8b) at the low frequency region indicates the ideal capacitive behavior where all the internal pores were fully infiltrated with the electrolyte ions [21]. KLCA88 showed more inclination compared to other KL-based Cas, indicating that the low surface area induced low ion accessibility into the electrodes which is in accordance with BET results (Figure 5). Ragone plots for KLCA-SCs are shown in Figure 8c and the sample KLCA60 performed better than other samples with an energy density of 5.67 Wh kg^−1^ at a power density of 50 W kg^−1^ and retained 3.36 Wh kg^−1^ at higher power density of 500 W kg^−1^. CVs and for GCD curves SLCA-SCs are shown in Appendix A, respectively. Because of the lower SSA and lack of hierarchical pore structure, SLCA-SCs exhibited poor electrochemical performances compared to KLCA-SCs. Figure 8d shows the comparison of specific capacitances of KLCA60 electrode with already reported carbon-based SCs from different precursor materials and it is clear that these CAs perform better than many of the CAs and carbon-based materials which are already reported in the literature [9,22,26,37,39,57,58]. In addition, cyclic stability studies performed using KLCA60 electrodes showed that the capacitance was retained by 90.2% after 2000 charge–discharge cycles, as shown in Appendix A, which confirmed the suitability of these materials as potential supercapacitor electrodes for energy storage applications.

The mechanical properties of CAs (KLCA60, KLCA88 and SLCA88) were analyzed in both the radial and axial directions and the stress–strain curves under compression are represented in Figure 8e,f, respectively. The axial compressive moduli were found to be higher than the radial ones for the tested carbon aerogels. For KLCA60, the values for axial and radial compressive moduli were 323 and 102 kPa, respectively. KLCA88 (420 and 170 kPa in axial and radial directions, respectively) and SLCA88 (622 and 136 kPa in axial and radial directions, respectively) showed higher moduli values compared to KLCA60 in both directions. These values agree with the lignin-based CAs in the literature where the longitudinal and transverse compressive moduli for lignin-reduced graphene oxide (RGO) foams were estimated to be 427 and 130 kPa, respectively [47]. In another study, Xu et al. reported a Young’s modulus of 0.328 MPa for bacterial cellulose/lignin carbon aerogels [22]. It was also observed that modulus values are depending on how dense the cell walls of CAs which in turn is related to their lignin contents as reported by Xu et al. [22]. Additionally, with the increase in lignin content, the brittleness of the CAs increased; hence, KLCA60 is more mechanically stable and handleable as SC electrodes compared to others.

## 4. Conclusions

A comparative study of electrochemical performances of green, extremely light, free standing, three dimensional (3D), anisotropic carbon aerogels (CAs) made using kraft (KL) and soda (SL) lignins along with cellulose nanofibers has been presented. Kraft lignin-based CAs showed remarkably high SSAs and porosity with numerous micro-, meso- and macroporous channels compared to soda lignin-based CAs. The clear differences between kraft- and soda lignin-based CAs throw light on the influence of lignin’s structural complexity, feedstock dependence and isolation process on the properties of CAs. By delineating the amount of CNFs, which acted as sacrificial templates during the carbonization process, the structure, morphology and hence the final electrochemical performances of CAs were tuned to attain high-performance supercapacitor (SC) electrodes. Among the different SCs studied here, KLCA60, with a specific surface area of 436 m^2^ g^−1^ and average pore diameter of 2.3 nm, showed a superior electrochemical performance with a specific capacitance of 163 F g^−1^ at a current density of 0.1 A g^−1^. An energy density of 5.67 Wh Kg^−1^ has also been achieved at a power density of 50 W Kg^−1^. Evidently, these 100% renewable porous carbon aerogel with excellent electrochemical properties are highly desirable candidates for supercapacitor electrode materials because they are tunable to achieve desired shape, structure, and morphology.

## Figures and Tables

**Figure 1 nanomaterials-11-00653-f001:**
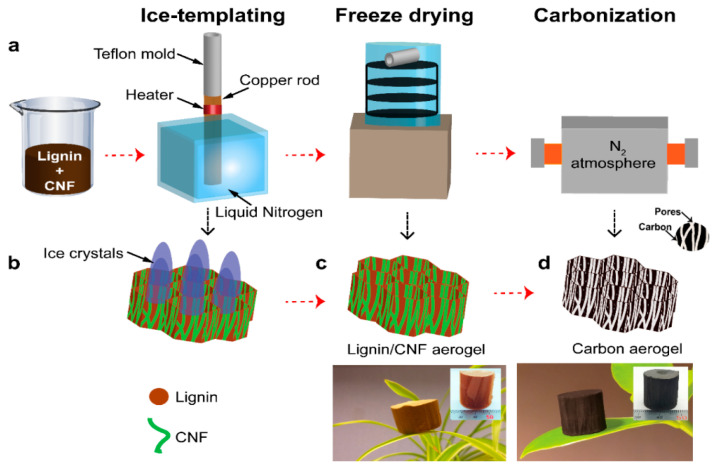
(**a**) Schematic showing the preparation of lignin-based CAs. Preparation of lignin/cellulose nanofiber (CNF) suspension, ice-templating process, freeze drying followed by carbonization in horizontal tubular furnace. Schematic representation of major events happening during the process; (**b**) unidirectional formation of ice crystals during ice-templating; (**c**) freeze-dried lignin/CNF aerogel with CNF incorporated lignin cell walls; (**d**) honeycomb-like porous CAs obtained by the sacrificial templating of CNF during carbonization.

**Figure 2 nanomaterials-11-00653-f002:**
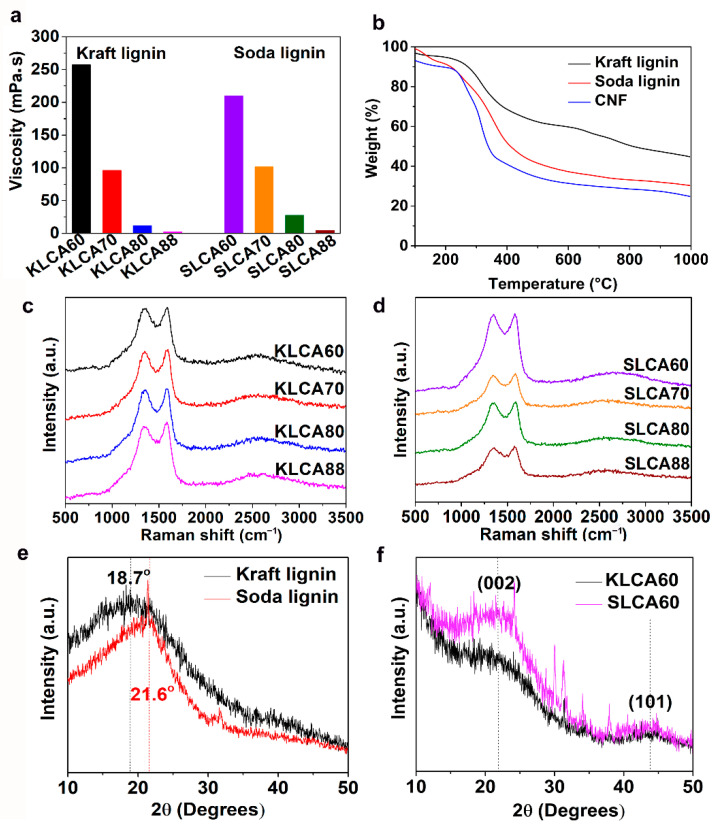
(**a**) Variation of viscosity with lignin concentration in lignin/CNF suspensions; (**b**) residue obtained as char at 1000 °C after the thermogravimetric analysis (TGA) of Kraft lignin (KL), soda lignin (SL) and CNF under N_2_ atmosphere. Raman spectra of (**c**) Kraft lignin-based carbon aerogels (KLCAs) and (**d**) soda lignin-based carbon aerogels (SLCAs), respectively. (**e**) XRD patterns for kraft and soda lignins before carbonization. (**f**) XRD patterns for KLCA60 and SLCA60.

**Figure 3 nanomaterials-11-00653-f003:**
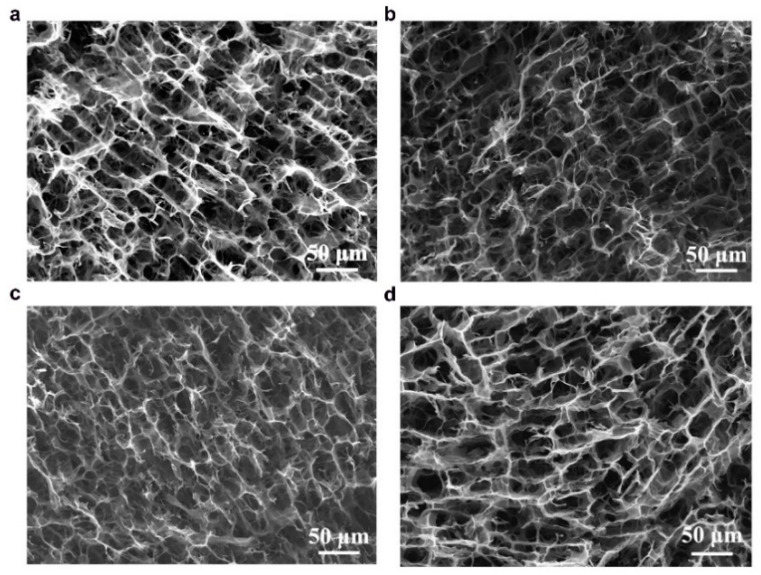
Honeycomb-like microstructures in the cross-section of CAs (**a**) KLCA60, (**b**) KLCA70, (**c**) KLCA80 and (**d**) KLCA88.

**Figure 4 nanomaterials-11-00653-f004:**
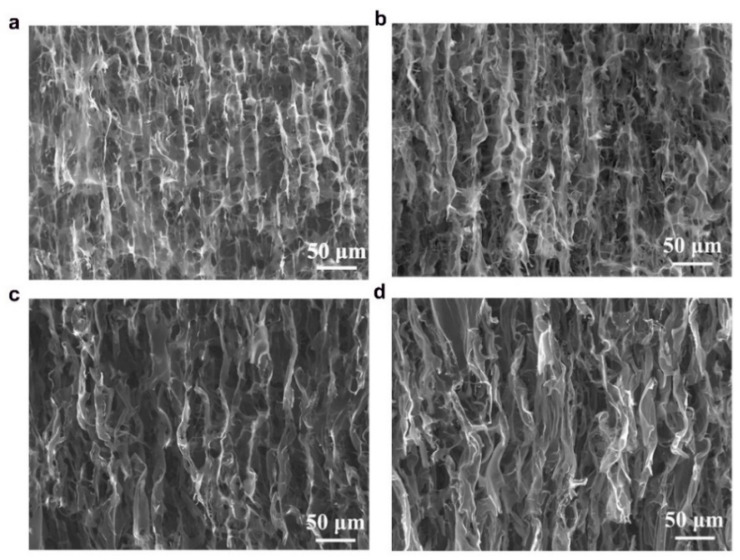
Longitudinal channel-like microstructure in the CAs (**a**) KLCA60, (**b**) KLCA70, (**c**) KLCA80 and (**d**) KLCA88.

**Figure 5 nanomaterials-11-00653-f005:**
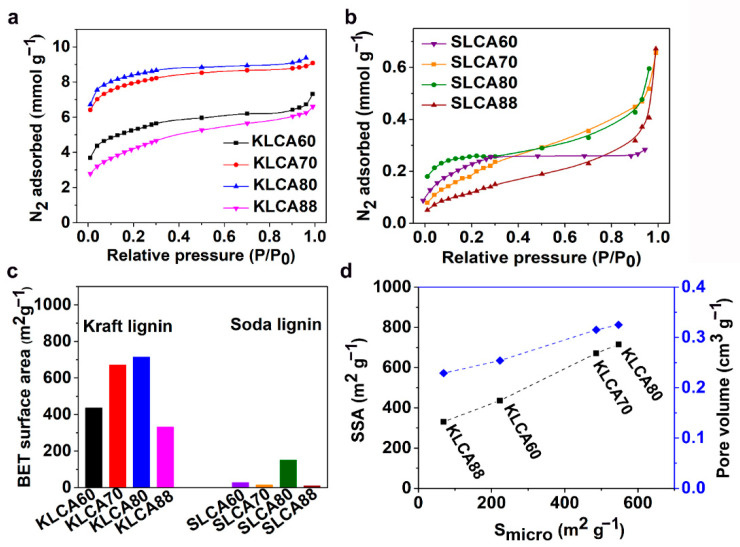
Results obtained from BET analysis: (**a**) nitrogen adsorption isotherms for KLCAs, (**b**) nitrogen adsorption isotherm for SLCAs, (**c**) BET surface area for KL- and SL-based CAs, and (**d**) variation of SSA with respect to the amount of micropores in KLCAs and relationship between total pore volume and micropore area in KLCAs.

**Figure 6 nanomaterials-11-00653-f006:**
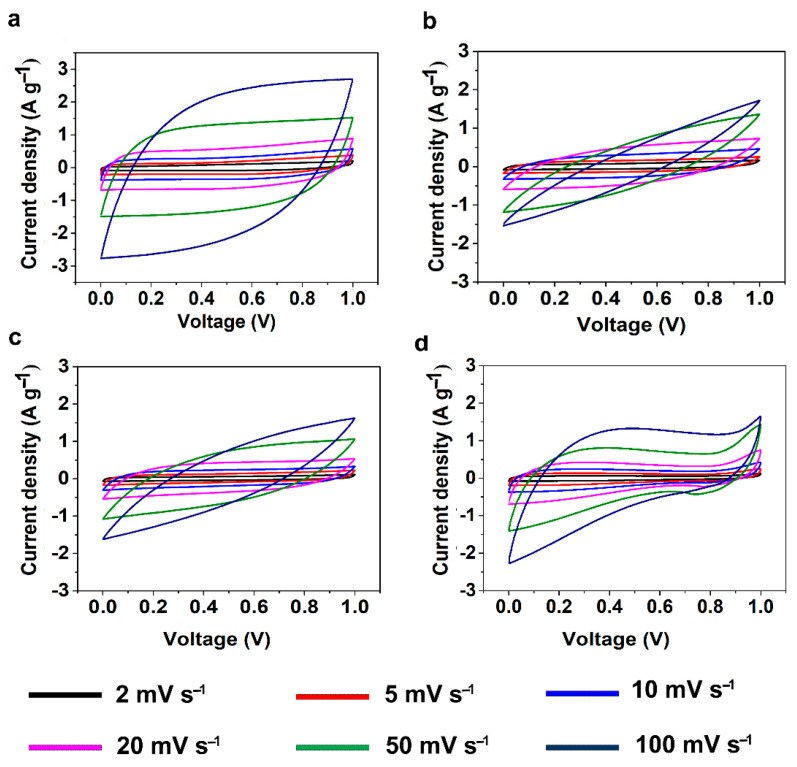
Cyclic voltammograms (CVs) for KLCA-supercapacitors (SCs) at different scan rates. (**a**) KLCA60, (**b**) KLCA70, (**c**) KLCA80 and (**d**) KLCA88 at 2, 5, 10, 20, 50 and 100 mV s^−1^.

**Figure 7 nanomaterials-11-00653-f007:**
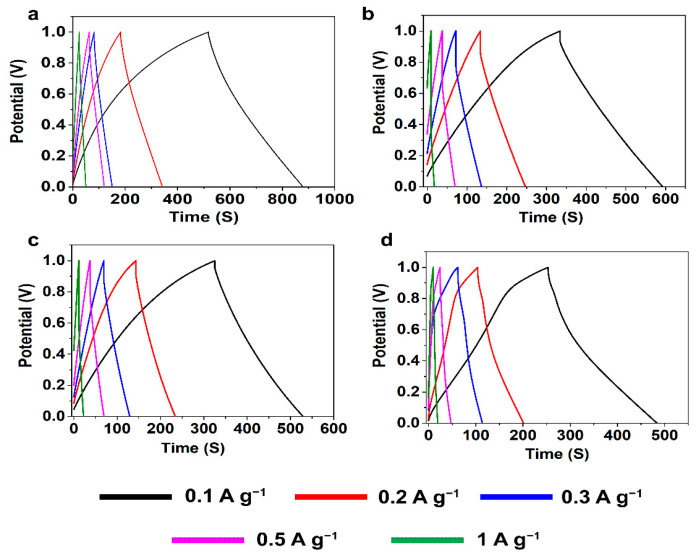
Galvanostatic charge discharge (GCD) curves for KLCA-SCs. (**a**) KLCA60, (**b**) KLCA70, (**c**) KLCA80 and (**d**) KLCA88 at different current densities of 0.1, 0.2, 0.3, 0.5 and 1 A g^−1^.

**Figure 8 nanomaterials-11-00653-f008:**
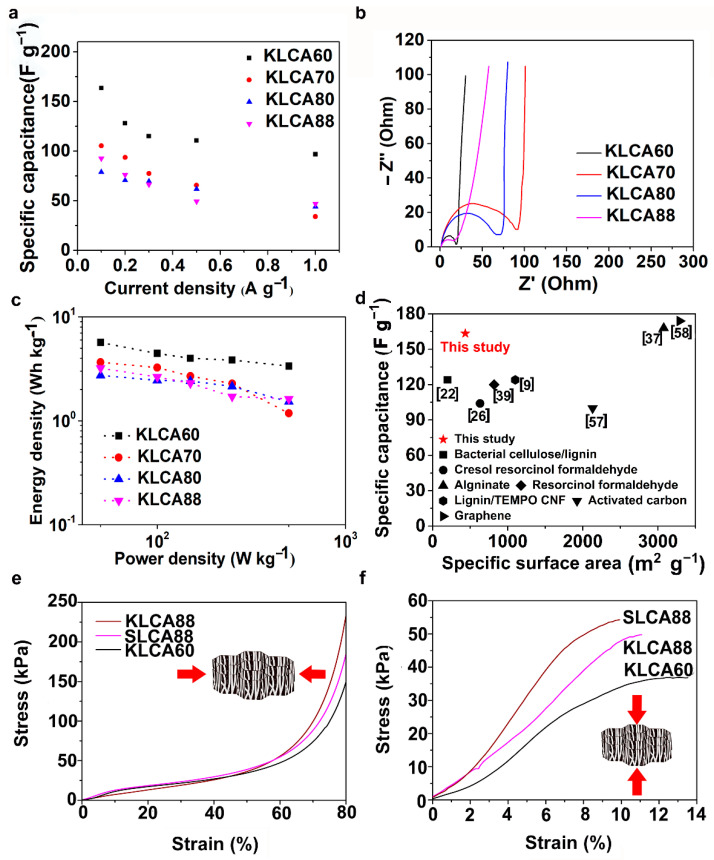
(**a**) Specific capacitance for KLCA-SCs at different current densities, (**b**) Nyquist plots, and (**c**) Ragone plots for the KLCA-SCs, and (**d**) comparison of specific capacitance of KLCA60 with CAs already reported from different precursor materials and the values are taken from references used in the study. Representative stress–strain curves for lignin-based CAs showing the response of KLCA88, SLCA88 and KLCA60 against compression in two directions—(**e**) radial direction and (**f**) axial direction.

**Table 1 nanomaterials-11-00653-t001:** Sample codes for the prepared carbon aerogels (CAs) and their compositions.

Sample Codes		Contents	
KL-based CAs	SL-based CAs	Lignin (%)	CNF (%)
KLCA60	SLCA60	60	40
KLCA70	SLCA70	70	30
KLCA80	SLCA80	80	20
KLCA88	SLCA88	88	12

**Table 2 nanomaterials-11-00653-t002:** Thermal properties of used materials, T_onset_, DTG_max_ and percentual amount of the residue, obtained from TGA.

	T_onset_ (°C)	DTG_max_ (°C)	Residue (%)
Kraft lignin	260	315	44
Soda lignin	161	357	30
CNFs	249	317	24

**Table 3 nanomaterials-11-00653-t003:** pH and conductivities for KL- and SL-based suspensions and physical properties such as bulk density, porosity, and volumetric shrinkage of CAs prepared using these suspensions.

Sample	Suspension Properties		Properties of Carbon Aerogels		
	pH	Conductivity (mS cm^−1^)	ρ_b_ (g cm^−3^)	P (%)	V_s_ (%)
KLCA60	8.89 ± 0.08	2.32 ± 0.02	0.046	97.95	66.65
KLCA70	9.05 ± 0.04	2.63 ± 0.02	0.042	98.16	67.12
KLCA80	9.11 ± 0.03	2.97 ± 0.03	0.052	97.71	69.97
KLCA88	9.17 ± 0.03	3.15 ± 0.01	0.061	97.3	71.09
SLCA60	9.10 ± 0.02	1.99 ± 0.01	0.024	98.92	69.19
SLCA70	9.49 ± 0.08	1.81 ± 0.06	0.026	98.84	70.94
SLCA80	9.23 ± 0.02	2.01 ± 0.01	0.028	98.77	72.23
SLCA88	8.56 ± 0.06	1.89 ± 0.01	0.033	98.54	73.26

ρ_b_—bulk density; P—porosity; V_s_—volumetric shrinkage.

**Table 4 nanomaterials-11-00653-t004:** Elemental composition in wt.% for KL- and SL-based carbon aerogels, obtained from scanning electron microscopy-energy dispersive X-ray spectroscopy (SEM-EDX) analysis.

Sample	C	O	Na	S	Si	K
KLCA60	75.80	10.20	7.70	7.63	-	-
KLCA70	74.89	13.04	6.85	5.19	-	-
KLCA80	84.43	9.80	3.68	1.79	-	-
KLCA88	88.50	7.98	2.56	0.93	-	-
SLCA60	87.40	7.70	3.20	1.40	0.30	-
SLCA70	84.01	6.32	4.96	2.50	1.20	0.90
SLCA80	83.94	7.97	3.91	2.16	1.17	0.84
SLCA88	86.30	8.30	3.50	1.80	-	-

**Table 5 nanomaterials-11-00653-t005:** Specific surface areas (SSAs) provided by micropores, contribution towards the SSA by meso- and macropores, average pore diameters and pore volumes obtained from Brunauer–Emmett–Teller (BET) surface area analysis.

Sample	SSA (m^2^ g^−1^)	S_micro_ (m^2^ g^−1^)	S_meso+macro_ (m^2^ g^−1^)	d_a_ (nm)	V_p_ (cm^3^ g^−1^)
KLCA60	436	223	213	2.30	0.254
KLCA70	671	486	185	1.88	0.315
KLCA80	715	547	168	1.82	0.325
KLCA88	331	69	262	2.75	0.229
SLCA60	28	7	21	2.02	0.014
SLCA70	14	-	14	6.37	0.023
SLCA80	151	82	69	2.20	0.083
SLCA88	10	-	10	9.39	0.023

SSA—specific surface area; S_micro_—micropore area; d_a_—average pore diameter; V_p_—pore volume.

## Data Availability

The data presented in this study is available on request from the corresponding author.

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
