# Peer review of "Hetero-Porous, High-Surface Area Green Carbon Aerogels for the Next-Generation Energy Storage Applications"

_nanomaterials, 2021, doi:10.3390/nano11030653_

Round 1

Reviewer 1 Report

This is an in-depth study on the structures and energy storage performance of carbon aerogels prepared from lignin and cellulose nanofiber precursors. The authors tuned the specific surface area, pore structure and therefore the specific capacitance of the carbon aerogels by changing the lignin to cellulose ratio in the precursors. I recommend the publication of this manuscript after solving a few minor issues.

 (1) Page 2, line 52. It would be better to provide bit more explanation on the energy recovery from lignin.

(2) Page 2, line 62, ‘low density and electrical conductivity’, low electrical conductivity is surely not favourable for electrode materials of supercapacitors.

(3) The abbreviations in the manuscript are not well defined and organized. For instance, KL, CL, TOCNTs, M-CNF, S/G ratio are not defined.

(4) The details regarding to the supercapacitor electrode and device fabrication should be provided in the experimental section. Did the author used the aerogel directly as an electrode without using binder and conductive additive (conventional powder composite electrode)? Were metallic current collectors used or not?

(5) From table 4, there are certain amount of Na and S in the samples, how would these elements affect the electrochemical performance? Can Na and S be removed by washing if they harm the electrochemical performance?

(6) Table 5, why does the specific surface area increase initially and then drop with the increase of lignin contents?

Author Response

Thank you very much for your positive and constructive feedback. We are grateful to have this opportunity to improve the quality of our manuscript accordingly, and we are glad to submit a revised version after addressing all the comments. The following text includes our point-by-point responses to the reviewer’s comments.

Responses to Reviewer #1:

Comments to the Author:

This is an in-depth study on the structures and energy storage performance of carbon aerogels prepared from lignin and cellulose nanofiber precursors. The authors tuned the specific surface area, pore structure and therefore the specific capacitance of the carbon aerogels by changing the lignin to cellulose ratio in the precursors. I recommend the publication of this manuscript after solving a few minor issues.

  • Page 2, line 52. It would be better to provide bit more explanation on the energy recovery from lignin.

Response: Thank you so much for the suggestion. Energy has been recovered from lignin in the form of heat, power, and biofuels. Line 52 has been modified including the ways in which energy recovery is done from various lignin resources.

(2) Page 2, line 62, ‘low density and electrical conductivity’, low electrical conductivity is surely not favourable for electrode materials of supercapacitors.

Response: We are thankful for the suggestion. This line has been changed to “low density and high electrical conductivity” in line 62 in the revised manuscript.

(3) The abbreviations in the manuscript are not well defined and organized. For instance, KL, CL, TOCNTs, M-CNF, S/G ratio are not defined.

Response: We are grateful for this comment. Corresponding changes have been made in the revised version of manuscript.

(4) The details regarding to the supercapacitor electrode and device fabrication should be provided in the experimental section. Did the author used the aerogel directly as an electrode without using binder and conductive additive (conventional powder composite electrode)? Were metallic current collectors used or not?

Response: Thank you for this relevant question. A new section has been added as ‘2.3 supercapacitor assembly’ in the experimental section to explain how the SC was assembled.

(5) From table 4, there are certain amount of Na and S in the samples, how would these elements affect the electrochemical performance? Can Na and S be removed by washing if they harm the electrochemical performance?

Response: We are grateful for the suggestion. Presence of Na and S could block some of the micro and mesopores in the carbon and the removal of these inorganics by washing could improve the specific surface area of carbon aerogels. However, we believe that the electrochemical performances will not show any variation from the current trend.

(6) Table 5, why does the specific surface area increase initially and then drop with the increase of lignin contents?

Response: Thank you for the question. Decrease in the specific surface area can be explained based on the microstructure of the carbon aerogels. In carbon aerogels prepared from lignin/CNF aerogels with high CNF content, specific surface area is derived from both sacrificial templating of CNF and also due to the evolution of volatiles during the carbonization of lignin. While those with high lignin content as in KLCA88, pores are mainly formed by the release of volatiles during the carbonization. In examining the longitudinal of microstructure of CAs, KLCA88 has the thickest and smoother cell walls which could lead to the trapping and deposition of volatiles evolved during carbonization of lignin. This trapping and condensation of volatiles could block the micro and meso pores and lead to the formation of denser carbon structure with lower specific surface area. The related explanation has been added to the revised manuscript.

Reviewer 2 Report

This work reports the preparation of carbon aerogels using kraft (KL) and soda (SL) lignins along with cellulose nanofibers. Clear differences between kraft and soda lignin based CAs were detected. CAs were used as high performance supercapacitor (SC) electrodes. The article may be interesting for the journal, but a thorough review must be carried out before publication. In order to increase the quality of the manuscript, authors should take into account the following considerations:

(1) In the description of the electrochemical analysis methods, the authors should indicate the cell typology (symmetric, asymmetric). What counter electrode was used in these tests?

(2) According to Figure 2b (TGA), KL presents a residue (mineral matter) higher than the SL sample. Therefore, looking at the profiles in the thermogram, the carbon yield of the KL sample should be noticeably lower than the SL sample. However, the carbon yields calculated in Figure S1b, and discussed on page 8 of the manuscript are similar for both carbonized samples (SLCA and KLCA). The authors should explain this discrepancy in the data.

(3) On page 9 the authors discuss the results of the EDX technique. However, this technique should never be used to discuss carbon content in samples, and even less so for properly carbonaceous samples. According to the results of the TG analysis, the content of mineral matter (residue) in the CAs obtained must be considerably high. Authors must present results for the actual carbon content in the KLCA and SLCA samples. This content must be confirmed by elemental analysis (CNHS) or by thermogravimetric analysis of these samples in an oxygen atmosphere (or air). The TG curves of all the aerogels obtained should be presented and discussed.

(4) In discussing the XRD patterns for the SLCA and KLCA samples, the authors do not discuss the clear signals of high crystallinity corresponding to the remaining mineral residues in these carbon aerogels. These signals, especially for the SLCA sample, must be assigned to their corresponding crystalline phases. If the calculations derived from the TG analysis are correct, the authors have obtained carbons with a high content of non-carbonaceous material, which is not described in the manuscript.

(5) The SEM photos presented in Figure 3 are too small and low resolution. A selection of photos with a larger size and high resolution should be selected, and the rest could be presented as a Supplementary Figure. The discussion about the microstructure cannot be verified with the current photos.

(6) The discussion of the porosimetry results (nitrogen adsorption) should be thoroughly reviewed: (a) First, the authors must provide a clear explanation for the anomalous textural properties of the KLCA88 sample (low SSA and pore volume), compared to the rest of the KLCA samples. Why is the increasing trend not maintained for that sequence of samples? (b) In table 5, the value of average pore diameter is indicated; however, this data is not relevant nor is it discussed in the text. (c) Usually, the pore volume is indicated as Vp. (d) Figures 4d and 4e do not provide relevant information: what conclusion is obtained from this graphic representation regarding the texture (porosity) of these materials? These are the same values ​​indicated in table 5 and in the text they are not interpreted regarding the KL content in each sample. (e) Authors should provide differential pore distribution curves and compare between porous samples (KLCA only).

(7) On the electrochemical properties, the authors should present the results of the KLCA electrodes in GCD long cycle tests. The stability of the capacitance throughout the cycles is a fundamental data to determine the quality of the designed supercapacitor.

(8) Surprisingly, the sample with the best capacitance results is the KLCA60, which does not show the highest values for specific surface area or pore volume. The authors propose that this fact “could be based from the perfect combination of approximately equal amounts of microporous (51.3% of total SSA) and meso / macro porous (48.7% of total SSA, contribution from macropores is very negligible since the average pore diameter is 2.3 nm) surface area in the material confirming the importance of achieving hierarchical pore morphology along with considerably high amount of mesoporous surface area. " This argument is unconvincing and not properly substantiated. The KLCA70 & 80 samples have practically the same SSAmeso as the KLCA60 sample, and also have a higher amount of micropores, which shortens the ion diffusion distance and thus improves the electrochemical performances of supercapacitors.  

Author Response

Thank you very much for your positive and constructive feedback. We are grateful to have this opportunity to improve the quality of our manuscript accordingly, and we are glad to submit a revised version after addressing all the comments. The following text includes our point-by-point responses to the reviewer’s comments

Responses to Reviewer #2:

Comments and Suggestions for Authors

This work reports the preparation of carbon aerogels using kraft (KL) and soda (SL) lignins along with cellulose nanofibers. Clear differences between kraft and soda lignin based CAs were detected. CAs were used as high performance supercapacitor (SC) electrodes. The article may be interesting for the journal, but a thorough review must be carried out before publication. In order to increase the quality of the manuscript, authors should take into account the following considerations:

(1) In the description of the electrochemical analysis methods, the authors should indicate the cell typology (symmetric, asymmetric). What counter electrode was used in these tests?

Response: Thank you for the comment. A new section has been added as ‘2.3 supercapacitor assembly’ in the experimental section to explain how the SC was assembled.

(2) According to Figure 2b (TGA), KL presents a residue (mineral matter) higher than the SL sample. Therefore, looking at the profiles in the thermogram, the carbon yield of the KL sample should be noticeably lower than the SL sample. However, the carbon yields calculated in Figure S1b, and discussed on page 8 of the manuscript are similar for both carbonized samples (SLCA and KLCA). The authors should explain this discrepancy in the data.

Response: We are grateful for the comment. As mentioned in the experimental section, the TGA analysis was done in N2 atmosphere which means that the final residue is the char obtained at 1000°C which consists of mostly the carbonaceous residue along with minor amounts of other inorganics which could be derived from the processing of lignin. Soda lignin used in this study is obtained from wheat straw, which contains additional inorganic elements like Si and K, which were confirmed qualitatively using EDX analysis as well. Also, we have added a small amount of 2 M sodium hydroxide solution to dissolve the soda lignin during the suspension preparation (see line 122-123). We observed higher tar formation and melting behavior of soda lignin that leads to condensation of volatiles in the CAs during the carbonization, which could also contribute to the higher yield for SLCAs. We are admitting the probability of misunderstanding the usage of ‘solid residue’ in the manuscript and hence we rephrased as ‘char residue’ to avoid any confusions in future.  

(3) On page 9 the authors discuss the results of the EDX technique. However, this technique should never be used to discuss carbon content in samples, and even less so for properly carbonaceous samples. According to the results of the TG analysis, the content of mineral matter (residue) in the CAs obtained must be considerably high. Authors must present results for the actual carbon content in the KLCA and SLCA samples. This content must be confirmed by elemental analysis (CNHS) or by thermogravimetric analysis of these samples in an oxygen atmosphere (or air). The TG curves of all the aerogels obtained should be presented and discussed.

Response: Thank you for your comment. EDX has been used by many researchers as a qualitative method to analyze the elemental composition of CAs [1–3]. As suggested, TGA in air atmosphere has been performed for KLCA60 and SLCA60 and result is included in the supporting information (Figure S4). KLCA60 showed only 6.9 % of ash at 1000°C and SLCA60 showed 3 % of ash. These results are consistent with the EDX analysis where KLCA60 has around 7 wt.% of metallic sodium which could remain as ash in air atmosphere. Similarly, SLCA60 showed the presence of Na, and Si (3.23 % total) in EDX analysis which can remain as metal oxides and silicates in the ash obtained in TGA. These results indicate that the CAs mainly consist of carbon as indicated in the EDX analysis.

(4) In discussing the XRD patterns for the SLCA and KLCA samples, the authors do not discuss the clear signals of high crystallinity corresponding to the remaining mineral residues in these carbon aerogels. These signals, especially for the SLCA sample, must be assigned to their corresponding crystalline phases. If the calculations derived from the TG analysis are correct, the authors have obtained carbons with a high content of non-carbonaceous material, which is not described in the manuscript.

Response: Thank you for the comment. From TGA analysis of KLCA60 and SLCA60 in air atmosphere, the mineral matter content is very less in the CAs and hence that perspective of analysis is irrelevant for the current study. Also, the XRD was done to confirm the formation of carbon which is clear from the (002) and (100) reflections in Figure 2f.

(5) The SEM photos presented in Figure 3 are too small and low resolution. A selection of photos with a larger size and high resolution should be selected, and the rest could be presented as a Supplementary Figure. The discussion about the microstructure cannot be verified with the current photos.

Response: Thank you for the valuable suggestion. SEM images have been modified and all the cross sections of KLCAS are represented in Figure 3 and microstructure in longitudinal direction is given in Figure 4 so that the images are more visible for analysis. SEM images of SLCAs are provided in the supporting information.

(6) The discussion of the porosimetry results (nitrogen adsorption) should be thoroughly reviewed: (a) First, the authors must provide a clear explanation for the anomalous textural properties of the KLCA88 sample (low SSA and pore volume), compared to the rest of the KLCA samples. Why is the increasing trend not maintained for that sequence of samples? (b) In table 5, the value of average pore diameter is indicated; however, this data is not relevant nor is it discussed in the text. (c) Usually, the pore volume is indicated as Vp. (d) Figures 4d and 4e do not provide relevant information: what conclusion is obtained from this graphic representation regarding the texture (porosity) of these materials? These are the same values ​​indicated in table 5 and in the text they are not interpreted regarding the KL content in each sample. (e) Authors should provide differential pore distribution curves and compare between porous samples (KLCA only).

Response: Thank you for the question. Decrease in the specific surface area can be explained based on the microstructure of the carbon aerogels. In carbon aerogels prepared from lignin/CNF aerogels with high CNF content, specific surface area is derived from both sacrificial templating of CNF and also due to the evolution of volatiles during the carbonization of lignin. While those with high lignin content as in KLCA88, pores are mainly formed by the release of volatiles during the carbonization. In examining the longitudinal of microstructure of CAs, KLCA88 has the thickest and smoother cell walls which could lead to the trapping and deposition of volatiles evolved during carbonization of lignin. This trapping and condensation of volatiles could block the micro and meso pores and lead to the formation of denser carbon structure with lower specific surface area. The related explanation has been added to the revised manuscript. As expected, Pore volume and SSA were increasing linearly with the amount of lignin in the KLCAs (As shown in Figure 5d and 5e in the revised manuscript) except for KLCA88 whose exceptional behavior is explained already. Unfortunately, the BET equipment available in our lab is not capable for performing adsorption and desorption studies to obtain the differential pore distribution. However, we believe that the current BET results give the sufficient details for understanding the porous structure of CAs.   

(7) On the electrochemical properties, the authors should present the results of the KLCA electrodes in GCD long cycle tests. The stability of the capacitance throughout the cycles is a fundamental data to determine the quality of the designed supercapacitor.

Response: Thank you for the suggestion. As we used self-made and open-to-air two-electrode setup for the electrochemical test using KOH as electrolyte, it is difficult to run cycle stability test continuously for a long time because the electrolyte will dry out and the consistency of the system cannot be guarantee.  However, we understand that cyclic stability studies are important for determining the quality of supercapacitor, which will be explored in future with a three-electrode set up.

(8) Surprisingly, the sample with the best capacitance results is the KLCA60, which does not show the highest values for specific surface area or pore volume. The authors propose that this fact “could be based from the perfect combination of approximately equal amounts of microporous (51.3% of total SSA) and meso / macro porous (48.7% of total SSA, contribution from macropores is very negligible since the average pore diameter is 2.3 nm) surface area in the material confirming the importance of achieving hierarchical pore morphology along with considerably high amount of mesoporous surface area. " This argument is unconvincing and not properly substantiated. The KLCA70 & 80 samples have practically the same SSAmeso as the KLCA60 sample, and also have a higher amount of micropores, which shortens the ion diffusion distance and thus improves the electrochemical performances of supercapacitors.

Response: Thank you for the comment. It is interesting that KLCA60 showed highest electrochemical performance. It is not the value of SSA that determines electrochemical performance of CAs. Effective utilization of SSA occurs when the pores are accessible for adsorption of ions and interconnected to allow the diffusion. That means adequate micro to meso pore ratio and interconnectivity of the structure are important factor in determining the effective utilization of SSA electrochemically which is already reported by Xu et al. [4] in the carbon aerogels derived from bacterial cellulose and lignin. So KLCA60 has nearly equal amount of meso and micro pores with average pore diameter 2.3 nm showed most favorable microstructure for electrolyte ions to adsorb. It is also well known that the all the micropores may not be available for electrolytes to adsorb. KLCA70, KLCA80 showed high micropores but at the same time their electrochemical performances indicate that those pores were not effectively accessible for the electrolyte ions to adsorb. Also, from the longitudinal microstructure of KLCAs (in figure 4) KLCA60 has lowest distance between adjacent walls which is beneficial in making EDLCs compared to those having higher distance between cell walls like in KLCA88. We agree that micropores shortens the ion diffusion distance, but accessibility is also equally important. Hence, In CAs it is equally important to have high amount of accessible micropores and mesopores which are interconnected, to exhibit the best electrochemical performance.

References

1. Saha, D.; Li, Y.; Bi, Z.; Chen, J.; Keum, J.K.; Hensley, D.K.; Grappe, H.A.; Meyer, H.M.; Dai, S.; Paranthaman, M.P.; et al. Studies on supercapacitor electrode material from activated lignin-derived mesoporous carbon. Langmuir 2014, 30, 900–910, doi:10.1021/la404112m.

2. Geng, S.; Wei, J.; Jonasson, S.; Hedlund, J.; Oksman, K. Multifunctional Carbon Aerogels with Hierarchical Anisotropic Structure Derived from Lignin and Cellulose Nanofibers for CO2 Capture and Energy Storage. ACS Appl. Mater. Interfaces 2020, 12, 7432–7441, doi:10.1021/acsami.9b19955.

3. Li, W.; Zhang, Y.; Das, L.; Wang, Y.; Li, M.; Wanninayake, N.; Pu, Y.; Kim, D.Y.; Cheng, Y.-T.; Ragauskas, A.J.; et al. Linking lignin source with structural and electrochemical properties of lignin-derived carbon materials. RSC Adv. 2018, 8, 38721–38732, doi:10.1039/C8RA08539K.

4. Xu, X.; Zhou, J.; Nagaraju, D.H.; Jiang, L.; Marinov, V.R.; Lubineau, G.; Alshareef, H.N.; Oh, M. Flexible, highly graphitized carbon aerogels based on bacterial cellulose/lignin: Catalyst-free synthesis and its application in energy storage devices. Adv. Funct. Mater. 2015, 25, 3193–3202, doi:10.1002/adfm.201500538.

Round 2

Reviewer 2 Report

The authors have responded positively to comments 1-5 from this reviewer. The explanations are scientifically correct and the modifications made are in accordance with the suggestions. However, two points remain to be clarified and must be addressed by the authors for the publication of the article.

(6) The differences in the textural properties of the KL and SL samples are crucial for their electrochemical performance. Therefore, the pore distribution curve should be provided by the authors, in order to provide significant differences between the samples. In response, the authors state: "Unfortunately, the BET equipment available in our lab is not capable for performing adsorption and desorption studies to obtain the differential pore distribution." However, to obtain the pore distribution curve, the authors do not need to develop new additional measurements. They should only apply the DFT or BJH models to the data presented in Figure 5a and 5b.

(7) Sorry but I must insist on the comment about the electrochemical properties. The authors should present the results of the KLCA electrodes in GCD long cycle tests. High capacitance values at various current densities are positive, but the stability of the electrode material during cycling is crucial in determining whether the material is suitable for supercapacitor applications.
